# The Texture Change of Chinese Traditional Pig Trotter with Soy Sauce during Stewing Processing: Based on a Thermal Degradation Model of Collagen Fibers

**DOI:** 10.3390/foods11121772

**Published:** 2022-06-16

**Authors:** Yuhai Lin, Ying Wang, Guofeng Jin, Junjie Duan, Yuemei Zhang, Jinxuan Cao

**Affiliations:** 1Hormel (China) Investment Co., Ltd., Jiaxing 314001, China; rubenlin@hormel.com.cn (Y.L.); patduan@hormel.com.cn (J.D.); 2School of Food and Health, Beijing Technology and Business University, Beijing 100048, China; wangying4@nbu.edu.cn (Y.W.); jgf@mail.hzau.edu.cn (G.J.); zhangyuemei@btbu.edu.cn (Y.Z.); 3College of Food and Biological Engineering, Chengdu University, Chengdu 610106, China

**Keywords:** pig trotter stewed with soy sauce, texture, stewing time, collagen fibers, secondary structures

## Abstract

In order to clarify the influence of the thermal degradation of collagen fibers on the texture profile analysis (TPA) parameters of pig trotter stewed with soy sauce (PTSWSS), TPA (springiness, chewiness, hardness, and gumminess), the secondary structures, the cross-linkage, decorin (DCN) and glycosaminoglycan (GAG) levels, and the histochemical morphology of collagen fibers during the stewing process (0, 30, 60, 120 min) were assessed. The springiness and hardness increased after 30 min of stewing, along with the denaturation of collagen proteins. TPA parameters improved with the prolonged stewing times of 60 and 120 min, along with the ultra-structural dissolution of collagen fibers, and a substantial reduction in cross-linkage, DCN, and GAG levels, and the unfolded triple-helix structure. This study concluded that the TPA parameters of PTSWSS were dependent on the stewing time, and that the improvement in TPA parameters with longer stewing time could primarily be attributed to the thermal degradation of collagen fibers.

## 1. Introduction

Pig trotter stewed with soy sauce (PTSWSS) is a very popular meat product in Asia, especially in China and Korea, due to its unique texture, taste, and flavor, and because it contains a large amount of collagen [1,2,3]. Chinese traditional PTSWSS is produced by washing, boiling in water for 5 min, cooling, and then stewing in marinated sauce for a long time. The main edible tissues of pig trotters are skins and tendons, which primarily consist of collagen type I [4]. Although some Western and European customers classify it as “abnormal food”, Chinese people believe that pig trotter has the potential to improve human health. Furthermore, the intake of collagen peptides from pig trotter with soy sauce during the stewing process has been demonstrated to decrease fragmentation of the collagen network [5]. In the Chinese market, the prices of pig trotters that are processed from connective tissues such as skin and tendons are 1.2–3.5 times those of “normal” meat products. A texture with a certain springiness, hardness, chewiness, and gumminess is considered an important sensory characteristic and quality attribute for product palatability, which determines the purchasing behavior of consumers [2].

Stewing, as a common cooking approach to preparing fresh meat products, could effectively improve the texture of meat products [6]. During stewing, meat proteins that are mainly composed of collagen and myofibrillar proteins denature [7]. The myofibril structure changes progressively, showing a shortening and condensing of I bands, and a slight stretching of A bands [8,9]. With a prolonged heating time, M and Z lines gradually become difficult to distinguish. In addition, the increase of total collagen and soluble collagen content in muscle tissue with the rise of thermal temperature and heating time has a great effect on the instrumental texture profile analysis [10]. However, the effect of stewing time on the texture of pig trotters comprised of collagen tissue is not well known.

Collagen is the most abundant protein in mammals and a minor component of connective tissues [4,11]. Collagen plays a significant role in the tensile strength of ligaments and tendons, in the elasticity of skin, and provides structural support for the cornea [4]. The right-handed triple helix collagen molecular structure is composed of three parallel polypeptide chains with 1300–1700 amino acid residues and the repeated G-X-Y sequence known as (G-X-Y)n, where G usually represents glycine, and X and Y often correspond to proline and hydroxyproline [12]. The collagen molecules were stabilized by the extensive hydrogen bonds of hydroxyproline, combined to form nanometer microfibrils, polymerized to form collagen fibrils, and further bundled into macroscopic collagen fibers [13]. Decorin (DCN), a chondroitin sulfate or dermatan sulfate proteoglycan, which is cross-linked with collagen to form collagen fibers, has a direct impact on the tensile properties of fibers [14,15]. Several studies on the effect of glycosaminoglycans (GAGs) on collagen fibrillogenesis have been performed [16]. Under thermal treatments, GAGs and DCN are not stable, and they can influence the precipitation kinetics and assembly of collagen fibrils [14,17]. However, the effect of stewing treatments on the structure and mechanical properties of collagen fibrils in pig trotters by modifying GAGs and DCN is far from understood.

Based on these considerations, the objective of this study is to clarify the influence of stewing time on the cross-linking degree, DCN and GAG content, the secondary structure and histochemical morphology of collagen fibers, and the texture profile of PTSWSS.

## 2. Materials and Methods

### 2.1. Procedures for Pig Trotter Stewed with Soy Sauce

A total of 60 pieces of raw pig trotters with an average approximate weight of 0.23–0.31 kg from Duroc × Landrance cross-breed pigs, slaughtered at the marketing weight of 102 ± 6 kg with an average age of 155 ± 15 d, were purchased from a local processing plant. Then, they were washed by tap water after the toes were cut off. After, the materials were cooked for 5 min in boiled brine (*v*/*w*, 200%) (including 3.3% rice wine, 7.3% salsa soya, 2.08% soy sauce, 12.5% fresh gingers, 20.8% onions, 4.17% salt (*w*/*w*)), washed with water, and cooled at room temperature for 1 h. Pig trotters were divided into 4 groups (15 pieces of pig trotter for each group) and were then marinated in boiled mixtures (*v*/*w*, 400%) (containing 2.7% bean sauce, 0.42% bunge prickly ash, 4.6% cinnamon, 1.25% star anise, 0.21% laurel leaf, 0.21% borneol, 0.42% amomums, 0.21% cloves, 0.42% pericarpium citri reticulatae, 0.21% Kaempferia galanga, 0.21% angelica dahuricas, 0.17% fennel, 0.1% white peppers, 0.42% galangas, 5.63% sugar, 4.2% broad bean paste, 2.29% soy sauce, 1.25% sodium glutamate, 2.2% salt (*w*/*w*)), followed by stewing separately at 95 °C for 0, 30, 60, and 120 min, respectively, in 4 H H-4 type thermostat water bath systems (Aohua Instrument Co., Ltd., Changzhou, Jiangsu, China).

### 2.2. Texture Profile Analysis (TPA) Measurements

For the convenience of the texture analysis, a rectangular cut of pig skin dermis (about 2 cm × 1 cm × 3.0 mm) was taken by surgical scissors and double-edge blade from PTSWSS after the visible hair was removed. All samples were measured by using a Texture Analyzer (Model, TA. XT; Make, Stable micro systems, Guildford, UK), which was equipped with a load cell of 1 kg and a cylindrical probe (P/5). In each case, the intact tissue was placed in the center of the object stage and compressed by 40% of its original height, with a rest period of 3 s between two cycles. The probe always returned to the trigger point before beginning the second cycle [18]. After the second cycle, the probe returned to its initial position. The pre-test speeds, test speeds and post-test speeds were 2 mm/s, 0.5 mm/s and 2 mm/s, respectively.

The measured parameters were springiness, hardness (g), chewiness, and gumminess. The data obtained from the TPA curve were used for the calculation of textural parameters [10]. Hardness is expressed as the maximum force for the first compression. Springiness was calculated as the ratio of time from the start of the second area up to the second probe reversal over the time between the start of the first area and the first probe reversal. Cohesiveness refers to a dimensionless parameter, meaning the ratio of the force peak area between the second and the first compression. Gumminess and chewiness have been reported as products of hardness and cohesiveness. Chewiness is represented as hardness multiplied by cohesiveness, multiplied by springiness. Gumminess is calculated as hardness multiplied by cohesiveness. All the measurements were taken at room temperature (25 °C) [19].

### 2.3. Histochemical Morphology

The histology analysis was done as described previously, with minor modifications [20]. For paraffin sections, pig skin tissues (about 0.5 cm × 0.5 cm × 0.5 cm) from PTSWSS were fixed by 4% parformaldehyde, dehydrated in a graded series (75% for 4 h, 85% for 2 h, 90% for 2 h, 95% for 1 h and 100% for 1 h) of ethanol, alcohol benzene (ethanol: xylene = 1:1, *v*/*v*) for 30 min, dimethylbenzene Ⅰ for 10 min, dimethylbenzene Ⅱ for 10 min, paraffin wax Ⅰ for 1 h, paraffin wax Ⅱ for 1 h and the paraffin wax Ⅲ for 1 h in a JJ-12J Dehydrator (Junjie Electronics Co., Ltd., Wuhan, Hubei, China) and embedded by the paraffin wax by using the JB-P5 Embedding machine (Junjie Electronics Co., Ltd., Wuhan, Hubei, China); 3 μm sections were cut parallel to collagen fiber direction and stained, with routine procedures used [21]. Sections were dewaxed in xylene Ⅰ and xylene Ⅱ for 20 min, respectively. Then, sections were hydrated in a graded series (100% ethanol Ⅰ for 10 min, 100% ethanol Ⅱ for 10 min, 95% for 5 min, 90% for 5 min, 80% for 5 min and 70% for 5 min) of ethanol and deionized water for 5 min. Subsequently, sections were stained with hematoxylin (Aladdin Biochemical Technology Co., Ltd., Shanghai, China) and eosin (Aladdin Biochemical Technology Co., Ltd., Shanghai, China), dehydrated in a graded series (95% Ⅰ, 95% Ⅱ, 100% Ⅰ, 100% Ⅱ) of ethanol for 5 min, respectively, marinated in xylene (Ⅰ, Ⅱ) for 5 min, respectively, and sealed in neutral gums. The histomorphological appearance of collagen fibers was observed by Nikon Eclipse Ti-SR inverted fluorescence microscope (Nikon, Tokyo, Japan).

### 2.4. Raman Spectroscopy

The collagen fiber of skin tissue from PTSWSS was measured using a Raman analyzer (Renishaw in Via Reflex Raman spectrometer; Renishaw, Gloucestershire, UK) according to a previous study [22]. The grounded and freeze-dried skin tissue samples were placed on microscope slides. An Argon Ion Laser that emitted at a wavelength of 785 nm was used as the excitation source. The scattered radiation was collected at 180° to the source; the typical spectra were recorded at 1 cm^−1^ resolution with 500–2100 cm^−1^ scans with 12 mW of laser power. The Phe m-ring band located near 1003 cm^−1^ was used as internal standard for normalization of the spectra. Eventually, the relative content of collagen secondary structures (α-helix, β-sheet, β-turn, random coil) was calculated by Peak Fit 4.12 (Sea Solve Software Inc., San Jose, CA, USA) according to our previous methods [23].

### 2.5. Determination of Cross-Links Degree and GAGs Content

The degree of cross-linkage and the content of total collagen and GAGs were measured according to the previous reports [24,25]. The samples were processed as follows: 1 g of skin tissue from the dermis layer from PTSWSS for every sample was homogenized by DY89-I high speed homogenizer (Scientz co., Ningbo, Zhejiang, China) in 10 mL of PBS (0.02 M, pH 7.2) on ice. Subsequently, the homogenized solution was centrifuged with a refrigerated centrifuge (Hunan Xiangyi, Laboratory Instrument Development Co., Changsha, China) at 1500× *g* for 15 min at 4 °C. Finally, the supernatant was collected for the determination of cross-linkage degree, total collagen and GAGs content with enzyme-linked immunoassay kits, respectively. A cross-links Elisa assay kit (E07P0743, Blue Gene Biotechnology Co., Ltd., Shanghai, China) with a pre-embedded mouse anti-cross-links monoclonal primary antibody, a total collagen Elisa assay kit (E07T0752, Blue Gene Biotechnology Co., Ltd., Shanghai, China) with a pre-embedded mouse anti-collagen monoclonal primary antibody (Genscript Biotechnology Co., Ltd., Nanjing, China) and a GAGs Elisa assay kit (E07G0301, Blue Gene Biotechnology Co., Ltd., Shanghai, China) with a pre-embedded mouse anti-GAGs polyclonal primary antibody (Genscript Biotechnology Co., Ltd., Nanjing, China), and corresponding HRP conjugated rabbit anti-mouse secondary antibodies (Genscript Biotechnology Co., Ltd., Nanjing, China) were used to determine the cross-linkage degree, total collagen, and GAGs content, respectively, according to manufacturer instructions. The results were expressed in μmol of pyridinoline, cross-links per g of collagen (μmol/g), and mg of GAGs per g of total collagen (mg/g).

### 2.6. Determination DCN Content

The DCN content was determined with a porcine DCN Elisa assay kit (Yuanye Biotechnology Co., Ltd., Shanghai, China) according to previous study [25]. Approximately 1 g of skin tissue samples from PTSWSS in five replicates without visible fat was homogenized with 10 mL of the phosphate buffer saline (1M, pH = 7.4) at 10,000 rpm for 30 s (homogenizing per 10 s with an interval of 15 s) on ice by a DY89-I high speed homogenizer (Scientz co., Ningbo, Zhejiang, China). The samples were centrifuged at 1000× *g* for 20 min at 4 °C with a refrigerated centrifuge (Xiangyi, Laboratory Instrument Development Co., Changsha, Hunan, China). The supernatant was used to determinate the DCN content by a porcine DCN Elisa assay kit (YY91980, Yuanye Biotechnology Co., Ltd., Shanghai China) with a pre-embedded mouse monoclonal primary antibody and a HRP-conjugated sheep secondary antibody (Genscript Biotechnology Co., Ltd., Nanjing, China) according to manufacturer instructions. 3,3’,5,5’-Tetramethylbenzidine (Aladdin Biochemical Technology Co., Ltd., Shanghai, China) as a substrate was converted to blue under the catalysis of peroxidase and converted to the final yellow color by the action of an acid. The depth of the color was positively correlated with the content of DCN in the sample. The absorption was measured at 450 nm with a 96-Well Plate Reader M200 (Tecan, Austria) to calculate the concentration. The results were expressed in μg of DCN per g of collagen (μg/g).

### 2.7. Statistical Analysis

Each pig trotter was represented at every treatment, resulting in 15 replicates per group. Duplicates were performed for each trotter in all measurements. All data of figures and tables were presented as the mean ± standard error. The effect of stewing time on the cross-links, GAGs, DCN, and secondary structure contents of collagen and the TPA of PTSWSS was analyzed via one-way analysis of variance procedure; means were compared using Duncan’s multiple range test of SAS 8.0 software (SAS Institute Inc., Cary, NC, USA). All the statistical significance of differences were set as *p* < 0.05.

## 3. Results and Discussion

### 3.1. The Influence of Stewing Time on the TPA Parameters

The TPA parameters (springiness, chewiness, hardness, and gumminess) of PTSWSS with different stewing times were shown in Figure 1. It was found that stewing time had a significant effect on TPA properties. Springiness increased from 0 to 30 min, while springiness (*p* < 0.01) and chewiness (*p* < 0.01) decreased from 30 min to 60 min. Hardness increased from 0 to 30 min, while hardness (*p* < 0.05) and gumminess (*p* < 0.05) decreased from 30 min to 60 min, and from 60 min to 120 min. It showed that stewing for 30 min increased springiness and hardness, while stewing for 60 and 120 min decreased springiness, hardness, chewiness, and gumminess.

The increase of springiness and hardness by stewing for 30 min could be attributed to the thermal denaturation of collagen proteins. This was very similar to a previous study [26], which demonstrated that the breaking strength of the collagen from bovine muscle increased at weak thermal treatments, but decreased at stronger thermal treatments, up to 90 °C. The decrease of TPA parameters from 30 min to 60 min shows an improvement in all the texture characteristics, while the further decrease of hardness and gumminess from 60 min to 120 min shows that stewing PTSWSS for a longer time leads to a product more suitable for the elderly, with improved chewing properties [27]. The cooking temperature has been recently been proven to produce an effect on the springiness and hardness of braised sauce porcine skin, with the highest temperature of 95 °C producing better product texture [28].

### 3.2. The Influence of Stewing Time on the Ultra Structure of Subcutaneous Fat

Adipose cell changes during stewing are displayed in Figure 2. The ultra structure of adipose cells showed significant breakage and fusion during 60 and 90 min of stewing, due to the indistinct boundary of the adipocyte membrane, according to the morphology compared to the control, while there was no obvious development at 30 min of stewing. Our results showed that prolonged stewing time accelerated adipose cell disruption.

Recent studies have shown that intramuscular fat content is closely connected with the eating quality of pork as related to meat tenderness [29]. We assume that adipose cell disruption might produce a beneficial effect on the chewiness and gumminess, thus contributing to meat tenderness.

### 3.3. The Influence of Stewing Time on the Ultra Structure of Collagen Fibers

The changes in collagen fibers by Van Gieson (ponceau) staining are shown in Figure 3. The ultra structure of collagen fibers for the control and samples at 30 min of stewing showed a very clear and classic fiber outline and intervals among fibers, while those samples at the 60 and 120 min of stewing showed an obscure and swollen overall appearance, and a significant disruption and fusion of collagen fibers. We assumed that a prolonged stewing time resulted in a depolymerization of collagen fibers and the dissolution of collagen proteins. Compared to samples at 60 min of stewing, those at 120 min of stewing were more obscure, with shorter intervals among fibers.

These results were very similar to those of a previous study [30], which indicated that swelling of collagen bands, thickening of collagen-rich layers, hyaline degeneration, and loss of birefringence were related to increased heating times. Xia et al. [31] observed the denaturation of fibril structures via scanning electron microscopy, and saw that fibrils of collagen became thicker and rougher after thermal treatments. The addition of salt has been reported to modify the structure of collagen [32], and at low concentrations, salt was observed to bind to the collagens, contributing to thermal transitions at a lower temperature [33].

The disruption and depolymerization of collagen fibers and the dissolution of collagen proteins during stewing for 60 and 120 min should be attributed to the improved texture. Collagen structure has been reported in our recent study to be destroyed when the cooking temperature was over 85 °C [28]. Also a 15 min heat treatment of turkey tendons at 95 °C was observed to solubilize collagen fragments [34]. Powell et al. [35] demonstrated that the solubilized collagen during heating treatment decreased Warner–Bratzler shear forces in bovine semitendinosus muscle. Bertola, Bevilacqua, and Zaritzky [36] reported that the hardness of semitendinosus muscle decreased with cooking time until reaching the lowest asymptotic values, due to the denaturation of collagen proteins within a cooking temperature range [36]. It was also observed that the solubility of the collagen content in fish muscle during storage time was closely linked to the firmness of sea bream muscle [37]. In our results, the disruption and depolymerization of collagen fibers weakened the mechanical strength and resistance of PTSWSS and caused a decrease in springiness, chewiness, hardness, and gumminess.

### 3.4. The Influence of Stewing Time on the Secondary Structure of Collagen Proteins

The α-helix of collagen proteins consists of a Gly-X-Y repeated structure, among which proline and hydroxyproline are the most common X and Y residues. Hydroxyproline tends to form hydrogen bonds via hydroxyl group cross-links, contributing to the stabilization of the helical structure of collagen [38]. Figure 4 presents the deconvolved and curve-fitted Raman bands of amide I (1600–1700 cm^−1^) in collagen proteins during stewing. It indicates that the frequencies of the amide I band components were correlated with the types of protein backbone conformation. The different peak regions have been assigned as secondary structural peaks [23,39]. The frequency and intensity changes in the Raman bands were the main indicators of changes in the secondary protein structure and of variations in the local environments of collagen. Our results showed that the intensities of α-helical peaks of stewing treatments at 30, 60 and 120 min were lower than those of the control, while the intensities of random coil peaks of stewing treatments at 30, 60 and 120 min were higher than those of the control. The levels of secondary structures of collagen proteins were shown in Table 1. At a stewing time of 30 min, a decrease of β-sheet and an increase of β-turn and random coil were shown. There was a significant decrease in α-helix (*p* < 0.01) and β-turn (*p* < 0.01) accompanied by a significant increase in random coil (*p* < 0.01) from 30 to 120 min during stewing. The β-sheet levels decreased from 30 to 60 min and then increased from 60 to 120 min (*p* < 0.05). These results of the transformation of α-helix and β-turn into random coil indicated that collagen proteins denatured and unfolded during the stewing process.

Xia et al. [31] assumed that heat-induced kinetic energy disrupted the secondary and tertiary structures, depending on the degree of thermal denaturation. Under moderate heating, only localized chain denaturing or breaking of a small number of β-sheet structures within collagen proteins occurred during 30 min of stewing. The denaturing increased the hardness and springiness of PTSWSS, but it had no influence on the ultra structure of collagen fibers. Under severe heating conditions during 60 and 120 min of stewing, however, the triple-helix-stabilizing hydrogen bonds were broken. As a result, a time-dependent irreversible transformation occurred, which led to the unfolding of the helices into a more random-coiled structure, the disruption of collagen fibers, and the dissolution of collagen proteins.

### 3.5. The Influence of Stewing Time on Cross-Links, DCN and GAGs Levels

Table 2 showed the cross-linkage, DCN and GAG levels of collagen during stewing. Compared to the control, these indicators significantly decreased during the entire stewing process. For pig trotters stewed for 0 min, the values of cross-links, DCN and GAGs were 56.78 μmol/g, 31.98 μg/g and 5.10 mg/g, respectively. The cross-linkage levels decreased by 36.23, 70.99 and 75.54% at 30, 60 and 120 min of stewing, respectively (*p* < 0.001). DCN levels decreased by 12.13, 70.82 and 82.33% at 30, 60 and 120 min of stewing, respectively (*p* < 0.001). GAG levels decreased by 28.82, 70.39 and 79.41% at 30, 60 and 120 min of stewing, respectively (*p* < 0.001). Our results showed that 60 min of stewing decreased the cross-linkage, DCN, and GAG values sharply, while 30 min of stewing only slightly weakened the cross-links, DCN, and GAGs of collagen.

In tissues, collagen is often created by interactions with other molecules to assemble the super-molecular collagen structure [40]. The cross-links of collagen networks were correlated with the mechanical properties of collagen tissues i.e., stretching and deformation abilities [41]. GAGs [42] and proteoglycans [43] were indicated to be key factors in the regulation of fibril assembly. DCN, an important proteoglycan with an average of 90–140 kDa, bridged neighboring type I collagen molecules to form collagen fibers, leaving the GAG chains to interact with one another and to form intermolecular visco-elastic bridges while strengthening the tensile properties of the fibers [14]. DCN also bound to other collagens, including types II, III, VI, and XIV, thereby playing a role in the matrix assembly [44]. Our recent study has proved that tendon samples showed higher thermal stability and GAG levels compared to those in skin tissues [45].

The results in this study were very similar to previous reports, which demonstrated that GAGs and DCN were not stable under thermal treatments [14,17]. Stewing for 60 min decreased the cross-links, DCN, and GAGs of collagen in PTSWSS sharply. As a result, it caused the looseness of the intermolecular viscoelastic bridges of collagen fibrils and exposed the triple-helix-stabilizing hydrogen bonds. In conclusion, the decrease in the cross-linkage, DCN, and GAG levels after being stewed for 60 min explained the unfolding of the secondary structure and indirectly contributed to the disruption of collagen fibers and the dissolution of collagen proteins. However, the nominal decrease of the cross-linkage, DCN and GAG levels after stewing for 30 min failed to expose the triple-helix-stabilizing hydrogen bonds.

## 4. Conclusions

The texture parameters (springiness, chewiness, hardness, and gumminess) of PTSWSS were dependent on stewing time. The increase of springiness and hardness at 30 min of stewing could be explained by the intra-molecular denaturation of collagen proteins, while the improvement of texture at the prolonged 60 and 120 min stewing times resulted from the substantial reduction in cross-linkage, DCN, and GAG levels, and from the unfolded triple-helix structure. The values of chewiness and gumminess decreased when stewed for 60 and 120 min, probably due to the disruption of the adipose cells and collagen fibers when stewed for a long time.

## Figures and Tables

**Figure 1 foods-11-01772-f001:**
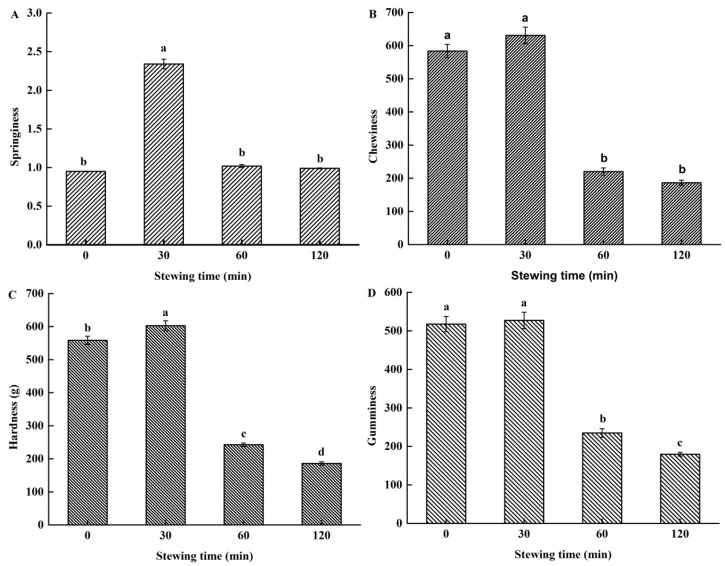
The effect of stewing time (0, 30, 60 and 120 min) on the texture. (**A**) for springiness; (**B**) for hardness (g); (**C**) for chewiness; (**D**) for gumminess. All values are means ± SE. ^a–d^ Different letters indicate significant differences among treatments (*p* < 0.05).

**Figure 2 foods-11-01772-f002:**
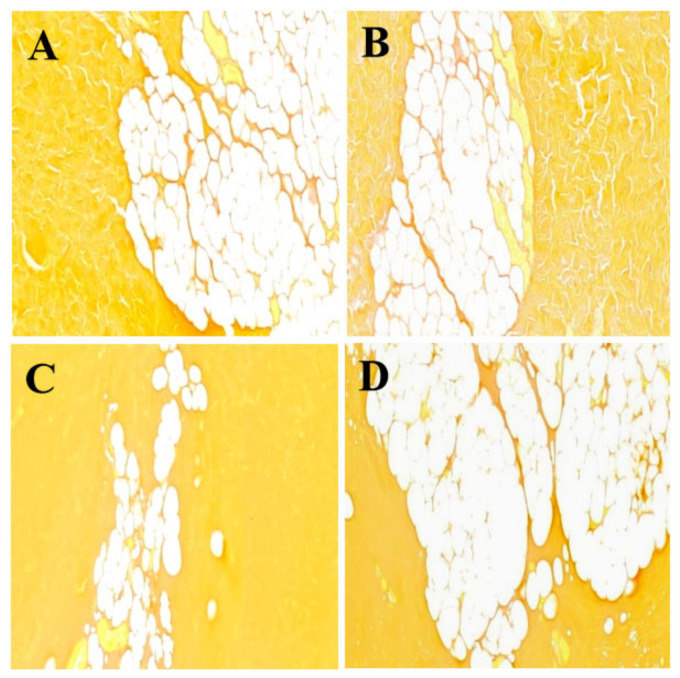
The effect of stewing time (0, 30, 60 and 120 min) on the ultra structure of fat globules. (**A**–**D**) represents samples at 0, 30, 60 and 120 min, respectively; magnification ×100.

**Figure 3 foods-11-01772-f003:**
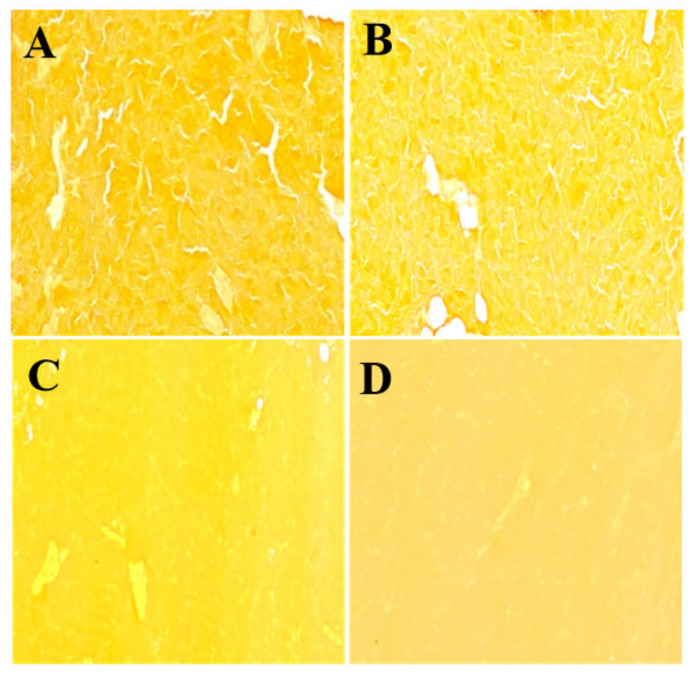
The effect of stewing time (0, 30, 60 and 120 min) on the ultra structure of collagen fibers. (**A**–**D**) represents samples at 0, 30, 60 and 120 min, respectively; magnification ×100.

**Figure 4 foods-11-01772-f004:**
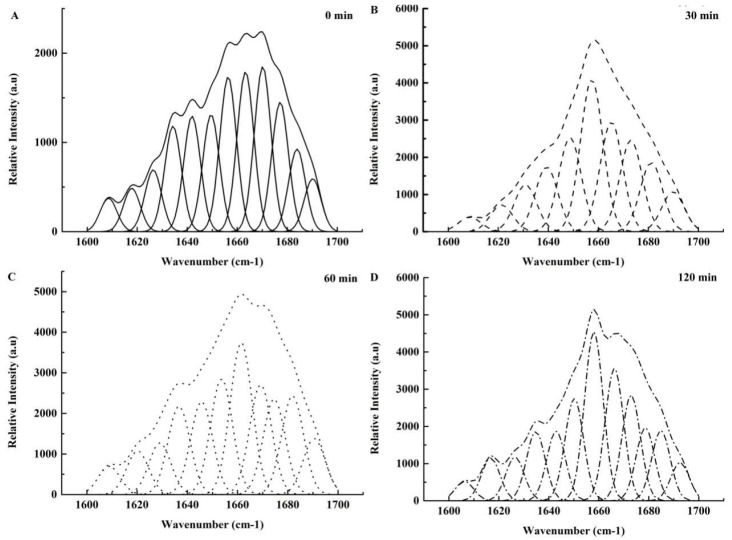
The effect of stewing time on the Raman spectra of collagen. (**A**–**D**) represents samples at 0, 30, 60 and 120 min, respectively.

**Table 1 foods-11-01772-t001:** The effect of stewing time (0, 30, 60 and 120 min) on the secondary structure of collagen.

Secondary Structure (%)	Stewing Time			
	0 min	30 min	60 min	120 min
α-helix	36.9 ± 0.6 a	35.1 ± 0.9 a	30.0 ± 1.2 b	27.5 ± 0.8 b
β-sheet	33.1 ± 0.6 a	24.8 ± 0.7 c	21.1 ± 0.7 d	28.2 ± 1.1 b
β-turn	20.4 ± 1.0 b	26.1 ± 2.0 a	25.5 ± 1.1 a	18.6 ± 1.2 b
Random coil	9.6 ± 0.5 c	14.0 ± 1.6 b	23.3 ± 0.9 a	23.4 ± 1.0 a

Different letters within columns and lines indicate significant differences among treatments (*p* < 0.05).

**Table 2 foods-11-01772-t002:** The effect of stewing time (0, 30, 60 and 120 min) on the cross-links, decorin and glycosaminoglycan (GAGs) of collagen.

Indicators	Stewing Time			
	0 min	30 min	60 min	120 min
Cross-links (μmol/g collagen)	56.8 ± 1.5 a	36.2 ± 3.0 b	16.5 ± 1.7 c	13.9 ± 0.4 c
Decorin (μg/g collagen)	31.98 ± 0.70 a	28.10 ± 0.61 b	9.33 ± 0.12 c	5.65 ± 0.10 d
GAGs (mg/g collagen)	5.10 ± 0.08 a	3.63 ± 0.01 b	1.51 ± 0.01 c	1.05 ± 0.02 d

Different letters within column and lines indicate significant differences among treatments (*p* < 0.05).

## Data Availability

Data is contained within the article.

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
