# Peer review of "The Texture Change of Chinese Traditional Pig Trotter with Soy Sauce during Stewing Processing: Based on a Thermal Degradation Model of Collagen Fibers"

_foods, 2022, doi:10.3390/foods11121772_

Round 1

Reviewer 1 Report

Dear authors,

Your study is dealing with national specialty, but the actual focus is more scientific.  The design of the study is very concise, and the study is well executed.  The methodology is very fine as well in most cases.  I think that the results can be interpreted for wider use than for pig trotters.  I would also wish that you will use the same sophisticated approach of yours to study other connective tissue aspects, like. tenderness in general and e.g. for woody breast problematics.

I have, however, some minor aspects to comment.

What is the slaughter age of the pigs?  This is of some relevance  when studying thew maturity of connective tissue.

L41-42.  The myofibrils structure changes progressively, showing that I bands shortened and condensed, and that bands appeared slightly stretched.  I am not sure that I understand this right.  First, I would write myofibril instead of myofibrils. Then, when you say that I bands are shortened and condensed, you say also that they are stretched.  Meaning I bands or other bands, and if I bands. how can they be then stretched?

L43.  “become indistinct gradually”   gradually become difficult to distinguish.

L96-104.  Should you also define cohesiveness?

L174.  “Replicated samples”?  This tells me that you have 15 sample units of each variable, and each sample has replicates.  How many replicates you did from each individual sample items, or do you only mean that that you had 15 replicates (or samples)?  Actually, ‘sample’ is a difficult word, sometimes it means one unit of a larger particle (and several samples are then replicates), sometimes a larger number of research units of the studied target group.

Chapter 2.1.  I am not sure about the sample treatments.  First they were boiled (the temperature must have been higher than  100 C, if the solution is really boiling)  for 5 minutes, then cooled, then stewed in 95 C (for how long?) AFTER? (having ?) being boiled 0 … 120 min) in water thermostat?  Actually these seem to mean stewing times, which is evident in later parts of the report, and the second mentioning of the boiling treatment may be unnecessary.  The process turns unclear because of the long list of ingredients of the brine/marinade. What were the pH values of the brine and marinade?  Please make this clearer for those who are not familiar with the procedure.

Chapter 2.2.

The sample was 20 x 10 x 3 mm.  I suppose that it was pressed against the shortest dimension (3 mm).  This was then pressed 1.2 mm.  How the sample was?  There were the outermost layers of skin from stratum corneum excluded (stratum lucidum, granulosum, spinosum, basale, about 0.3 mm thick) and then dermis (corium).  If the sample was then 3 mm thick, did you check how much of subcutaneous fatty tissue with loose connective tissue was included?

L207-208.  Is this reference really relevant?  Fish fats are more unsaturated than pork and the connective tissue also more sensitive to cooking.  Andersen et al. (2015) only cited, not studied themselves?  The citation is also given twice in the same sentence;  should be (Andersen et al., 2015).

L211-212.  “cell”  cells.  You assume that the lubrication improves (present tense) in mouth and decreased (past tense) chewiness and gumminess.  You did not do sensory analyses?  I do not like the combination these parameters in this way.  Please rephrase.

To be honest, Fig. 2 and Fig. 3 do not look very good and informative: small micrographs and two times almost the same thing.  The reader cannot easily reach the same conclusions as you do just by looking the micrographs.  Please reconsider.

L228-244.  Did you find any studies about effects of salt on pig skin or collagen in general?

The scientifically most important part of the results is in the chapters 3.4 and 3.5.  This is most interestinf indeed!

As a small detail: I do not particularly like the use of two decimals in your tables.  As you can see, the variation of your second decimals is even 296 times (Table 2, cross-links, 30 min.).  Two decimals are not needed in Table 1, but only with decorin and GAGs in Table 2.

 L294 and 296.  Remove the initials from the references.

L321.  What does improving of gumminess mean?  An increase or decrease?

List of reference.  I do not know the practices of this journal, but it looks odd that the references are numbered and not in alphabetical order: they should be numbered in the text, or in alphabelical order in the list of refrerences.

Author Response

Your study is dealing with national specialty, but the actual focus is more scientific.  The design of the study is very concise, and the study is well executed.  The methodology is very fine as well in most cases.  I think that the results can be interpreted for wider use than for pig trotters.  I would also wish that you will use the same sophisticated approach of yours to study other connective tissue aspects, like. tenderness in general and e.g. for woody breast problematics.

Thank you for your good suggestions concerning our manuscript. We will consider it in the future study.

I have, however, some minor aspects to comment.

What is the slaughter age of the pigs?  This is of some relevance when studying thew maturity of connective tissue.

The Duroc × Landrance cross-breed pigs were slaughtered with an average age of 155 ± 15 d, and this has been clarified in the manuscript.

L41-42.  The myofibrils structure changes progressively, showing that I bands shortened and condensed, and that bands appeared slightly stretched.  I am not sure that I understand this right.  First, I would write myofibril instead of myofibrils. Then, when you say that I bands are shortened and condensed, you say also that they are stretched.  Meaning I bands or other bands, and if I bands. how can they be then stretched?

Thanks for your suggestion. “I bands shortened and condensed and the A bands appeared slightly stretched”. This is an error in writing and has been corrected in the manuscript.

L43.  “become indistinct gradually” à  gradually become difficult to distinguish.

 Thanks for your suggestion. We revised it accordingly.

L96-104.  Should you also define cohesiveness?

 Thanks for your suggestion. We revised it accordingly.

L174.  “Replicated samples”?  This tells me that you have 15 sample units of each variable, and each sample has replicates.  How many replicates you did from each individual sample items, or do you only mean that that you had 15 replicates (or samples)?  Actually, ‘sample’ is a difficult word, sometimes it means one unit of a larger particle (and several samples are then replicates), sometimes a larger number of research units of the studied target group.

“Each pig trotter was represented at every treatment resulting in 15 replicates per group. Duplicates were performed for each trotter in all measurements”. This has been clarified in the manuscript.

Chapter 2.1.  I am not sure about the sample treatments.  First they were boiled (the temperature must have been higher than 100 C, if the solution is really boiling)  for 5 minutes, then cooled, then stewed in 95 C (for how long?) AFTER? (having ?) being boiled 0 … 120 min) in water thermostat?  Actually these seem to mean stewing times, which is evident in later parts of the report, and the second mentioning of the boiling treatment may be unnecessary.  The process turns unclear because of the long list of ingredients of the brine/marinade. What were the pH values of the brine and marinade?  Please make this clearer for those who are not familiar with the procedure.

Thanks for your suggestion. This has been clarified in the manuscript. “Pig trotters were divided into 4 groups (15 pieces of pig trotter for each group) and then were marinated in boiled mixtures (v/w, 400%) (comprising 2.7% bean sauce, 0.42% bunge prickly ash, 4.6% cinnamons, 1.25% star anise, 0.21% myrcia, 0.21% borneol, 0.42% amomums, 0.21% cloves, 0.42% pericarpium citri reticulatae, 0.21% Kaempferia ga-langa, 0.21% angelica dahuricas, 0.17% fennel, 0.1% white peppers, 0.42% galangas, 5.63% sugar, 4.2% broad bean paste, 2.29% soy sauce, 1.25% aginomoto, 2.2% salt (w/w)), followed by stewing at 95 °C for 0, 30, 60, and 120 min, respectively, corresponding to different groups. A H H-4 type thermostat water bath system (Aohua Instrument Co., Ltd., Changzhou, Jiangsu, China) was used to boil the above marinates”.

Chapter 2.2.

The sample was 20 x 10 x 3 mm.  I suppose that it was pressed against the shortest dimension (3 mm).  This was then pressed 1.2 mm.  How the sample was?  There were the outermost layers of skin from stratum corneum excluded (stratum lucidum, granulosum, spinosum, basale, about 0.3 mm thick) and then dermis (corium).  If the sample was then 3 mm thick, did you check how much of subcutaneous fatty tissue with loose connective tissue was included?

Thanks for your suggestion. A rectangular shape of pig skin (about 2 cm × 1 cm × 3.0 mm) of dermis was taken by surgical scissors and double-edge blade from PTSWSS. We considered that the dermis layer decided the texture of Pig trotter, since it was the thickest skin among all skin layers and indeed contained subcutaneous fatty tissue and loose connective tissue. Unfortunately, we have not checked the amount of the subcutaneous fatty tissue in this study. This would certainly be done in the future study.

L207-208.  Is this reference really relevant?  Fish fats are more unsaturated than pork and the connective tissue also more sensitive to cooking.  Andersen et al. (2015) only cited, not studied themselves?  The citation is also given twice in the same sentence;  should be (Andersen et al., 2015).

Thanks for your suggestion. This reference has been deleted.

L211-212.  “cell” à cells.  You assume that the lubrication improves (present tense) in mouth and decreased (past tense) chewiness and gumminess.  You did not do sensory analyses?  I do not like the combination these parameters in this way.  Please rephrase.

Thanks for your suggestion. This has been rephrased in the manuscript.

To be honest, Fig. 2 and Fig. 3 do not look very good and informative: small micrographs and two times almost the same thing.  The reader cannot easily reach the same conclusions as you do just by looking the micrographs.  Please reconsider.

Thanks for your good suggestion. We have revised the figures as you wish.

L228-244.  Did you find any studies about effects of salt on pig skin or collagen in general?

Some references focusing on the effects of salt on collagen in general have been added in the manuscript.

The scientifically most important part of the results is in the chapters 3.4 and 3.5.  This is most interesting indeed!

Thanks for your good comments.

As a small detail: I do not particularly like the use of two decimals in your tables.  As you can see, the variation of your second decimals is even 296 times (Table 2, cross-links, 30 min.).  Two decimals are not needed in Table 1, but only with decorin and GAGs in Table 2.

Thanks for your suggestion. This has been modified in the tables.

 L294 and 296.  Remove the initials from the references.

Thanks for your suggestion. This has been modified.

L321.  What does improving of gumminess mean?  An increase or decrease?

“The values of chewiness and gumminess decreased stewed for 60 and 120 min, probably due to the disruption of the adipose cell at a long time of stewing”. This has been clarified in the conclusion.

List of reference.  I do not know the practices of this journal, but it looks odd that the references are numbered and not in alphabetical order: they should be numbered in the text, or in alphabelical order in the list of refrerences.

Thanks for your suggestion. The references have been numbered in the text.

Reviewer 2 Report

Manuscript ID: foods-1721299

In the article entitled: “The texture change of Chinese traditional pig trotter with soy  sauce during stewing processing: based on a thermal degradation model of collagen fibers” was study the influence of the thermal degradation of collagen fibers on the texture  parameters of pig trotter  for different times stewed with soy sauce. In addition, studies were carried out the secondary structures, the cross-links, decorin, glycosamino-glycans levels, and histochemical morphology of collagen fibers during the stewing process.

This article has been written in a compact manner. From the methodological point of view, the employed measurement techniques are appropriate to the adopted objective of the research work. The results obtained may have practical application.

 Title

The title and the aim of the study are clearly constructed.

Abstract

The abstract includes the aim of the study, methods used in the experiment and contain the principal results and conclusions.

Introduction

The introduction describes the matter of the experiment accurately and clearly states the problem being investigated.

Methods

The data is well collected. The methods are described in detail, in the way which permits the research to be replicated. The sampling is appropriate and adequately described.

Results

The results were discussed extensively, in a clear and legible way.

Discussion

They correctly interpreted and described the significance of the results for the research. They skillfully referred to the results of other researchers.

References

The references are accurate.

Language

The article is correctly written.

However, the work itself is not revealing. The impact of heating on the thermal degradation of collagen fibers and its relationship with muscle texture is well known. An interesting element of the work is the raw material used. Although it appears in many national cuisines, so far I have practically not met it in studies known to me.

Author Response

Thank you for your good suggestions concerning our manuscript. Pig trotter stewed with soy sauce is regarded as a very popular meat product in Asia, especially in China and Korea. Stewing of pig trotter with soy sauce is exclusively for the Chinese consumers. However, the effect of this cooking method on the structure and mechanical properties of collagen fibrils in pig trotters has been scarcely approached, although it would be highly appreciated for the industry and the consumers.

Reviewer 3 Report

Peer Review report 1 on: “The texture change of Chinese traditional pig trotter with soy sauce during stewing processing: based on a thermal degradation model of collagen fibers”.

  1. Original submission.
    • Recommendation

Major revisions

  1. Comments to Author

Manuscript Number: Foods-1721299

Title: The texture change of Chinese traditional pig trotter with soy sauce during stewing processing: based on a thermal degradation model of collagen fibers

Overview and general recommendation:

The results are interesting and may contribute to the understanding of the effect of stewing time on collagen composition and microstructure of skin and tendons/ligaments in a traditional food such as pig trotters. Nevertheless, my main concerns are; a) All treatments were boiled and stewed with the same brine. How can be separated the effect of the brines from the effect of the stewing time? From my personal point of view, it was necessary to add control groups with no brine, to conclude precisely if the changes in texture are by the time, the brine, or the combination. In this sense, I recommend adding information on the brines effect on the measured variables. B) My other major concern is the use of sodium cyanide in the formula. Is that correct? Is it safe and ethical to use it in food?

The section of materials and methods omits some information about the research, which may be useful for the discussion of the study and its repeatability. This section needs to be completed and described in detail.

The section of results and discussion must be improved. The discussion is not properly focused and incomplete. Additionally, the references used for discussion are not relevant or recent. 

  • Major comments

-Material and methods:

-Procedures for pig trotter stewed

Line 72. What is the age-weight of pigs commercially slaughtered in the plant? Is the animal age important to the chemical state of collagen fibres and texture?

Line 75. What is the difference between salsa soya and soy sauce? It seems a synonym to me.

Lines 78-82. Used the scientific names for plants or herbs when they are not popular (i.e. octagon, myrcia, galangal, aginomoto, etc.)

Line 80. Sodium cyanide? Is not that toxic-poisonous? (https://www.sciencedirect.com/topics/medicine-and-dentistry/sodium-cyanide)

-Texture profile

Line 87. ‘A rectangular shape of…’ instead of ‘the rectangular…”

Lines 99-103. It is not better to add equations instead of the description of calculations? Look less confusing with equations.

-Histochemical morphology

Add the brand and concentration of every reactive used in your study.

-Cross-link degree and GAGs content

Lines 146-147. Information of mouse anti-cross-links monoclonal primary antibody and HRP conjugated rabbit anti-mouse secondary antibodies?

Did you make the analysis only on the skin (Line 139)? What about tendons/ligaments which are abundant in trotters?

-Determination DCN content

Again, the analysis was only on the skin, what about tendons/ligaments? It is necessary to add the information of antibodies used, and chemicals such as methylbenzidine.

-Statistical analysis

How many DCN content, cross-link, GAGs analysis, spectroscopy, etc. were performed from every 15 trotters/treatment? Why the replicate is considered for every trotter? What was the experimental unit?  

-Results and discussion

-TPA parameters

I understand that samples in treatment 0 were only boiled for 5 min. What is the biochemical reason to be less hard than those boiled for 5 plus 30 stewed min? The discussion of the variables is focused on bovine muscle, but it would be better to focus it on tendons/ligaments with more recent references.

   -Ultra-structure of subcutaneous fat

What methodology/data was used to determine that adipose cells from control and 30 showed significant break and fusion in comparison to 60 and 90 (Lines 203-204)? Did the authors perform some analysis of the area, surface, distances, etc.? How to conclude that? There is anything described in the mat. and methods section, nor this section. How were selected the image sections appearing in the manuscript? There must be described a methodology leading to that result. 

  • Ultra-structure of collagen fibres

Homogenize: “ultra-structure” or “ultra structure”.

As in the previous section, it may be perceived that fibre outlines and intervals among them are reduced in 60- and 90-min treatments, Nevertheless, there should be a methodology to quantify and analyse if there is a statistical difference. The use of established methodologies avoids using assumptions to declare results or to make declarations based on perceptions.

Once again, the discussion is based on muscle, not in connective tissue (tendons) and the references are very old.

  • Secondary structure of collagen proteins

To describe the relationship of secondary structure with TPA is very adequate. However, the chemical bases need to be clarified. Discussion is limited, hence, it should be widened and pointing the effect of time and temperature applied in the experiment on the collagen.

  • Cross-link, DCN and GAGs levels

Values from raw trotters do not need to be listed in the text and in the Table. Remove them from the text.

If cross-links, DCN and GAGs are reduced since the 30 min treatment, why did springiness, hardness, and chewiness increase in that treatment?  It does not make sense, does it?

Line 295. Which mechanical properties?

Kalamajski et al. described those bonds by DCN in fibrillogenesis. Does that apply to the cooked tendons/tissue?

  • Minor comments

There are some misspellings along with the document. The language in the whole document must be re-checked and mistakes need to be corrected.

Line 235. Remove “Powell et al.” from brackets.

Line 268. Subtract “Xia et al” out of brackets.

Line 281. Delete the word “of” in “GAGS of were…”

Author Response

Overview and general recommendation:

The results are interesting and may contribute to the understanding of the effect of stewing time on collagen composition and microstructure of skin and tendons/ligaments in a traditional food such as pig trotters. Nevertheless, my main concerns are; a) All treatments were boiled and stewed with the same brine. How can be separated the effect of the brines from the effect of the stewing time? From my personal point of view, it was necessary to add control groups with no brine, to conclude precisely if the changes in texture are by the time, the brine, or the combination. In this sense, I recommend adding information on the brines effect on the measured variables. B) My other major concern is the use of sodium cyanide in the formula. Is that correct? Is it safe and ethical to use it in food?

Thank you for your good suggestions concerning our manuscript. This study is more focused on the changes of the thermal degradation of collagen involving structure and mechanical properties induced by the stewing processing rather than the brine during marination. All of treatments samples were followed by stewing separately at 95 °C for 0, 30, 60, and 120 min, respectively, in 4 thermostat water bath systems . So, we don’t need consider the effect of the brines from the effect of the stewing time on the texture. However, the addition of salt has been reported to modify the structure of collagen and some references focusing on the effects of salt on collagen in general have been added in the manuscript. We will consider this in the future study. Sodium cyanide in the formula is a written mistake and we have not used this substance in the study. This has been corrected in the manuscript.

The section of materials and methods omits some information about the research, which may be useful for the discussion of the study and its repeatability. This section needs to be completed and described in detail.

Thank you for your good suggestions. This has been modified in the materials and methods.

The section of results and discussion must be improved. The discussion is not properly focused and incomplete. Additionally, the references used for discussion are not relevant or recent. 

Thank you for your good suggestions. More recent references have been added in the manuscript.

Major comments

-Material and methods:

-Procedures for pig trotter stewed

Line 72. What is the age-weight of pigs commercially slaughtered in the plant? Is the animal age important to the chemical state of collagen fibres and texture?

Duroc × Landrance cross-breed pigs slaughtered at the marketing weight of 102 ± 6 kg with an average age of 155 ± 15 d were purchased from a local processing plant.

Line 75. What is the difference between salsa soya and soy sauce? It seems a synonym to me.

We do not know how to spell the two spices in English. By consulting the supermarket salesperson, we confirmed that they are bean sauce and broad bean paste.

Lines 78-82. Used the scientific names for plants or herbs when they are not popular (i.e. octagon, myrcia, galangal, aginomoto, etc.)

Thank you for your good suggestions. This has been modified in the manuscript.

Line 80. Sodium cyanide? Is not that toxic-poisonous? (https://www.sciencedirect.com/topics/medicine-and-dentistry/sodium-cyanide)

Sodium cyanide in the formula is a written mistake and we have not used this substance in the study. This has been corrected in the manuscript.

-Texture profile

Line 87. ‘A rectangular shape of…’ instead of ‘the rectangular…”

This has been modified in the manuscript.

Lines 99-103. It is not better to add equations instead of the description of calculations? Look less confusing with equations.

Thank you for your good suggestions. This has been modified in the manuscript.

-Histochemical morphology

Add the brand and concentration of every reactive used in your study.

Thank you for your good suggestions. This has been added in the manuscript.

-Cross-link degree and GAGs content

Lines 146-147. Information of mouse anti-cross-links monoclonal primary antibody and HRP conjugated rabbit anti-mouse secondary antibodies?

Thank you for your good suggestions. This has been added in the manuscript.

Did you make the analysis only on the skin (Line 139)? What about tendons/ligaments which are abundant in trotters?

-Determination DCN content

Again, the analysis was only on the skin, what about tendons/ligaments? It is necessary to add the information of antibodies used, and chemicals such as methylbenzidine.

Thank you for your good suggestions. In fact, we took 1 g of skin tissue samples of dermis by surgical scissors and double-edge blade from PTSWSS. We considered that the dermis layer decided the texture of Pig trotter, since it was the thickest skin among all skin layers. Unfortunately, we have not determined the tendons/ligaments tissue in this study. It will certainly be done in the future study.

In addition, the information of antibodies and methylbenzidine were added.

-Statistical analysis

How many DCN content, cross-link, GAGs analysis, spectroscopy, etc. were performed from every 15 trotters/treatment? Why the replicate is considered for every trotter? What was the experimental unit?  

Each pig trotter was represented at every treatment resulting in 15 replicates per group. Duplicates were performed for each trotter in all measurements. This has been clarified in the manuscript.

-Results and discussion

-TPA parameters

I understand that samples in treatment 0 were only boiled for 5 min. What is the biochemical reason to be less hard than those boiled for 5 plus 30 stewed min? The discussion of the variables is focused on bovine muscle, but it would be better to focus it on tendons/ligaments with more recent references.

Thank you for your good suggestions. The hardness value was higher in samples stewed for 30 mins as compared to non-stewing, which probably due to the thermal denaturation of collagen with increased time of stewing. More recent references have been added in the manuscript.

   -Ultra-structure of subcutaneous fat

What methodology/data was used to determine that adipose cells from control and 30 showed significant break and fusion in comparison to 60 and 90 (Lines 203-204)? Did the authors perform some analysis of the area, surface, distances, etc.? How to conclude that? There is anything described in the mat. and methods section, nor this section. How were selected the image sections appearing in the manuscript? There must be described a methodology leading to that result. 

Thank you for your suggestions. The ultra-structure of adipose cell showed significant break and fusion during 60 and 90 min stewing due to the indistinct boundary of adipocyte membrane according to the morphology compared to the control. This has been clarified in the manuscript.

  • Ultra-structure of collagen fibres

Homogenize: “ultra-structure” or “ultra structure”.

Thanks for your suggestions, and this has been modified.

As in the previous section, it may be perceived that fibre outlines and intervals among them are reduced in 60- and 90-min treatments, Nevertheless, there should be a methodology to quantify and analyse if there is a statistical difference. The use of established methodologies avoids using assumptions to declare results or to make declarations based on perceptions.

Thank you for your suggestions. Some representative samples from each treatment were selected for qualitative analysis. The change in morphology was observed and unfortunately we did not conduct the quantitative analysis for the ultra-structure of the collagen fibers.

Once again, the discussion is based on muscle, not in connective tissue (tendons) and the references are very old.

Thank you for your good suggestions. More recent references have been added in the manuscript.

  • Secondary structure of collagen proteins

To describe the relationship of secondary structure with TPA is very adequate. However, the chemical bases need to be clarified. Discussion is limited, hence, it should be widened and pointing the effect of time and temperature applied in the experiment on the collagen.

Thank you for your good suggestions. The chemical bases have been clarified and more recent references on the effects of time and temperature have been added in the manuscript.

  • Cross-link, DCN and GAGs levels

Values from raw trotters do not need to be listed in the text and in the Table. Remove them from the text.

Thanks for your suggestions. The pig trotters without stewing were set as the control group in this study aiming to show the thermal degradation of collagen fibers as increased with stewing time.

If cross-links, DCN and GAGs are reduced since the 30 min treatment, why did springiness, hardness, and chewiness increase in that treatment?  It does not make sense, does it?

There might be a delay occurred between the physicochemical modifications and the corresponding textural changes. The thermal degradation of the collagen fibres during stewing for the first 30 min was believed to cause reduced values of cross-links, DCN and GAGs. However, the increased textural values might be related to the thermal denaturation of the collagen proteins which seemed to produce a larger effect than that from the thermal degradation at the beginning of stewing.

Line 295. Which mechanical properties?

The properties of stretching and deformation and this has been added in the manuscript.

Kalamajski et al. described those bonds by DCN in fibrillogenesis. Does that apply to the cooked tendons/tissue?

DCN was described to be quite abundant in skin and tendon tissues and showed a high-affinity collagen-binding site in the leucine-rich repeats. We assumed that these bonds by DCN could be applied to the skin tissue during cooking.

  • Minor comments

There are some misspellings along with the document. The language in the whole document must be re-checked and mistakes need to be corrected.

Line 235. Remove “Powell et al.” from brackets.

Line 268. Subtract “Xia et al” out of brackets.

Line 281. Delete the word “of” in “GAGS of were…”

Thanks for your suggestions. This has been modified in the manuscript.